# RNA Virus Discovery Sheds Light on the Virome of a Major Vineyard Pest, the European Grapevine Moth (*Lobesia botrana*)

**DOI:** 10.3390/v17010095

**Published:** 2025-01-13

**Authors:** Humberto Debat, Sebastian Gomez-Talquenca, Nicolas Bejerman

**Affiliations:** 1Instituto de Patología Vegetal, Centro de Investigaciones Agropecuarias, Instituto Nacional de Tecnología Agropecuaria (IPAVE-CIAP-INTA), Camino 60 Cuadras Km 5,5, Córdoba X5020ICA, Argentina; 2Unidad de Fitopatología y Modelización Agrícola, Consejo Nacional de Investigaciones Científicas y Técnicas (UFYMA-CONICET), Camino 60 Cuadras Km 5,5, Córdoba X5020ICA, Argentina; 3Estación Experimental Agropecuaria Mendoza, Instituto Nacional de Tecnología Agropecuaria (EEA-Mendoza-INTA), San Martín 3853, Luján de Cuyo, Mendoza 5507, Argentina

**Keywords:** RNA virome, *Lobesia botrana*, European grapevine moth, vineyards, data mining

## Abstract

The European grapevine moth (*Lobesia botrana*) poses a significant threat to vineyards worldwide, causing extensive economic losses. While its ecological interactions and control strategies have been well studied, its associated viral diversity remains unexplored. Here, we employ high-throughput sequencing data mining to comprehensively characterize the *L. botrana* virome, revealing novel and diverse RNA viruses. We characterized four new viral members belonging to distinct families, with evolutionary cues of cypoviruses (*Reoviridae*), sobemo-like viruses (*Solemoviridae*), phasmaviruses (*Phasmaviridae*), and carmotetraviruses (*Carmotetraviridae*). Phylogenetic analysis of the cypoviruses places them within the genus in affinity with other moth viruses. The bi-segmented and highly divergent sobemo-like virus showed a distinctive evolutionary trajectory of its encoding proteins at the periphery of recently reported invertebrate Sobelivirales. Notably, the presence of a novel phasmavirus, typically associated with mosquitoes, expands the known host range and diversity of this family to moths. Furthermore, the identification of a carmotetravirus branching in the same cluster as the Providence virus, a lepidopteran virus which replicates in plants, raises questions regarding the biological significance of this moth virus to the grapevine host. We further explored viral sequences in several publicly available transcriptomic datasets of the moth, indicating potential prevalence across distinct conditions. These results underscore the existence of a complex virome within *L. botrana* and lay the foundation for future studies investigating the ecological roles, evolutionary dynamics, and potential biocontrol applications of these viruses in the *L. botrana*–vineyard ecosystem.

## 1. Introduction

The European grapevine moth (*Lobesia botrana*), a pest native to the Palearctic region, has become one of the most economically damaging insects for viticulture worldwide. Its larval stages cause direct damage to grape berries, facilitating the establishment of secondary fungal infections, such as *Botrytis cinerea*, which further compromise fruit quality and yield. This pest exerts significant economic pressure on grape growers, with estimated losses ranging from reduced yields to increased costs associated with pest management programs [1]. In Greece, for example, losses in grape production caused by *L. botrana* have been reported to range between 13.3% and 27% over a four-year period, underscoring its substantial impact on the viticultural industry [2].

Since its emergence in Europe, *L. botrana* has rapidly expanded its range, replacing or marginalizing other historically significant grape pests, such as *Eupoecilia ambiguella* and *Sparganothis pilleriana*, due to its adaptability and high reproductive potential [3]. In countries like Chile, Argentina, and the USA, its recent establishment has necessitated stringent quarantine measures, including fumigation with methyl bromide, which compromises fruit quality and increases management costs. The pest was first detected in America in 2008, establishing itself in Chile and Argentina and subsequently appearing in the United States in 2009, where it was reported in Napa Valley, CA, USA [1,4]. This expansion highlights the urgent need for sustainable, effective pest-control strategies.

Current management practices rely heavily on chemical control, mating disruption, and quarantine measures [5]. However, growing concerns over pesticide resistance, environmental impact, and regulatory restrictions on chemical use have prompted the development of alternative strategies. Among these, the sterile insect technique (SIT) has shown promise in urban and peri-urban areas, where conventional control methods are challenging [4]. Similarly, biological control approaches, such as the use of entomopathogenic fungi and parasitoids like *Goniozus legneri*, have demonstrated efficacy in reducing pest populations while minimizing environmental impact [6,7].

Despite extensive research on its ecology, behavior, and management strategies [1,5], a significant gap persists in our knowledge of the *L. botrana* virome. This knowledge gap hinders a comprehensive understanding of *L. botrana* population dynamics and its susceptibility to environmental stressors. Viral infection has demonstrable effects on arthropod biology, influencing fitness, behavior, and resistance to environmental changes [8,9]. In agricultural ecosystems, these interactions can profoundly impact productivity, pest dynamics, and the efficacy of control measures [10,11]. Consequently, uncovering the composition and diversity of the *L. botrana* virome is essential to understanding this plague and designing more effective pest management strategies.

In this line, we initiated here an extensive exploration of the *L. botrana* virome through the integration of high-throughput sequencing (HTS) outputs. This in silico strategy allowed us to surpass the limitations of traditional virus isolation techniques, capturing an unbiassed viral repertoire of invertebrates, including both known and potentially novel viral lineages [12,13]. Our investigation focused on characterizing the *L. botrana* virome to comprehensively define the taxonomic composition and diversity of novel viral lineages, unveiling the hidden viral counterpart shaping the moth’s biology. By shedding light on the previously unseen viral dimension of *L. botrana* biology, we aimed to provide valuable insights to foster future studies on the ecological roles of viruses within this significant agricultural pest. Additionally, a glimpse of the *L. botrana* virome contributes to a more comprehensive understanding of insect–virus interactions within agricultural ecosystems, potentially impacting broader ecological and evolutionary studies, valuable for the development of novel, virus-based control strategies. This study introduces a pivotal step towards comprehending a crucial facet of this economically important pest and contributing to a more sustainable future for vineyards.

## 2. Materials and Methods

### 2.1. Database Selection and High-Throughput Sequencing (HTS) Library Processing

To characterize the *L. botrana* virome, publicly available metatranscriptomic RNA-Seq datasets from *L. botrana* individuals collected from diverse geographic regions and carried out by [14,15] and Reineke et al. (PRJNA910346)were downloaded from the NCBI Sequence Read Archive (SRA). The datasets included in this work are available in the NCBI SRA archive at https://www.ncbi.nlm.nih.gov/sra/?term=txid209534[Organism:noexp] (accessed on 19 October 2024) and described in Table 1.

### 2.2. Bioinformatics Analysis, Sequence Assembly, and Virus Identification

Virus discovery procedures were conducted following established methodologies [16,17]. Briefly, raw nucleotide sequence reads from each *L. botrana* SRA experiment were retrieved from their respective NCBI BioProjects (Table 1). The datasets underwent preprocessing, including trimming and filtering, using the Trimmomatic v0.40 tool, accessed via http://www.usadellab.org/cms/?page=trimmomatic (accessed on 30 October 2024). Standard parameters were employed, apart from raising the quality required from 20 to 30 (initial ILLUMINACLIP step, sliding window trimming, average quality required = 30). The resulting reads were assembled de novo using rnaSPAdes with standard parameters on the Galaxy server (https://usegalaxy.org/), accessed on 30 October 2024. Subsequently, transcripts obtained from the de novo transcriptome assembly underwent bulk, local BLASTX searches (E-value < 1 × 10^−5^) against the complete NR release of viral protein sequences, which was retrieved from https://www.ncbi.nlm.nih.gov/protein/?term=txid10239[Organism:exp], accessed on 30 October 2024. The resulting viral sequence hits from each dataset were thoroughly examined. Tentative virus-like contigs were curated, extended, and/or confirmed through iterative mapping of filtered reads from each SRA library. This iterative strategy involved extracting a subset of reads related to the query contig, utilizing the retrieved reads to extend the contig, and repeating the process iteratively using the extended sequence as the new query. The extended and polished transcripts were subsequently reassembled using the Geneious v8.1.9 alignment tool (Biomatters Ltd., Boston, MA, USA) with high-sensitivity parameters.

### 2.3. Bioinformatics Characterization of Novel Viral Genomes

Open reading frames (ORFs) were predicted using ORFfinder with a minimal ORF length of 150 nt and genetic code 1 (https://www.ncbi.nlm.nih.gov/orffinder/, accessed on 30 October 2024). The functional domains and architecture of translated gene products were determined using InterPro (https://www.ebi.ac.uk/interpro/search/sequence-search, accessed on 30 October 2024) and the NCBI Conserved domain database-CDD v3.20 (https://www.ncbi.nlm.nih.gov/Structure/cdd/wrpsb.cgi, accessed on 30 October 2024) with an e-value threshold of 0.01. Additionally, HHPred and HHBlits, as implemented in https://toolkit.tuebingen.mpg.de/ (accessed on 30 October 2024), were employed to complement the annotation of divergent predicted proteins using hidden Markov models. Transmembrane domains were predicted using the TMHMM version 2.0 tool (http://www.cbs.dtu.dk/services/TMHMM/, accessed on 30 October 2024). The predicted proteins were then subjected to NCBI-BLASTP web searches against the non-redundant protein sequences (nr) database to filter out any endogenous virus-like sequences that did not show virus protein as the best hit. Phylogenetic analysis based on the predicted polymerase protein of all available viruses was carried out using MAFFT 7.526 (https://mafft.cbrc.jp/alignment/software/, accessed on 30 October 2024) with multiple aa sequence alignments using G-INS-i (LbSV, LbCaV) and E-INS-i (LbPV, LbCV) as the best-fit model, respectively. The aligned aa sequences were used as input to generate phylogenetic trees through the maximum-likelihood method with the FastTree 2.1.11 tool available at http://www.microbesonline.org/fasttree/ (accessed on 30 October 2024). Local support values were calculated with the Shimodaira–Hasegawa test (SH) and 1000 tree resamples. The polymerase proteins of Sin Nombre virus—NP_941976 and Thottapalayam virus—YP_001911124 (LbPV), Rice ragged stunt virus—AAC36456 (LbCV), Imperata yellow mottle virus—NC_011536, Southern cowpea mosaic virus—NC_001625, and Turnip rosette virus—NC_004553 (LbSV) were used as the outgroup in the phylogenetic trees.

## 3. Results

### 3.1. Virus Discovery by Data Mining of Metatranscriptomic RNA-Seq Libraries

To characterize the *L. botrana* virome, we analyzed existing RNA-Seq datasets deposited in the NCBI SRA from *L. botrana* individuals with diverse origins, biological characteristics, and host conditions. Reads were retrieved from 29 publicly available libraries and processed. Through similarity searches against a viral protein database using BLASTX searches, we identified contigs and ORFs homologous to sequences from known viruses. Four novel viruses were discovered, reconstructed, and further characterized.

### 3.2. Bioinformatics Characterization of Novel Viral Genomes

#### 3.2.1. A Novel Phasmavirus Linked to *L. botrana*

A novel phasmavirus was identified and named *Lobesia botrana* phasmavirus (LbPV). This virus was detected in two transcriptome datasets from pheromone glands sampled from adult individuals collected in Germany in 2016 (Table 1). The reconstructed LbPV genome (GenBank accession numbers BK067724-BK067726) comprised three segments (RNA1 = L, RNA2 = M and RNA3 = S) of single-stranded, negative-sense RNA of 6458 nt, 2518 nt, and 1759 nt, respectively (Figure 1A) with the 3′ end terminal sequence conserved among the viral segments, sharing the consensus sequence of other phasmaviruses such as Anopheles triannulatus orthophasmavirus (AtoPV, Figure 1E). Segment L has one ORF that encodes a putative RdRp protein of 2100 aa, segment M has one ORF that encodes a putative glycoprotein precursor of 766 aa (G), and segment S has two overlapping ORFs, where ORF1 encodes a putative non-structural protein (NSs) of 125 aa, while the ORF2 encodes a putative nucleoprotein (NP) of 384 aa (Figure 1A). Segment S was the most abundant one in terms of coverage, whereas segment M was the one that accumulated less on those datasets where LbPV was identified (Figure 1A). The RdRp protein shows the highest BlastP similarity with the moth-associated pink bollworm virus 2 (PBV2) RdRp, with 50.43% identity (Appendix A), and included a Bunya_RdRp superfamily conserved domain (8C4V_A, bunyavirus RdRP, Hantaan virus, E-value 8 × 10^−104^) identified in its sequence at aa positions 909–1237 (Figure 1A). All typical N-terminal domains, pre-motif, and motifs A-E present in the bunyavirids encoded RdRps were identified in LBPV (Figure 1B). The G protein showed the highest BlastP similarity with the moth-associated Seattle prectang virus (SEPV) G protein, with 32.63% identity (Appendix A) and an EnvGly conserved domain (4HJ1_A ENVELOPE GLYCOPROTEIN; Class II fusion protein E-value 1.3 × 10^−21^) was identified in its sequence at aa positions 392–466 (Figure 1A). The putative G precursor is predicted to be processed by a conserved signal peptide peptidase to yield two mature G proteins (Gn and Gc) (Figure 1A,D). The NP protein shows the highest BlastP similarity with the SEPV NP, with 43.56% identity (Appendix A), and no conserved domains were identified in its sequence. The putative NSs have no hits in the BlastP searches, and no conserved domain was identified in the sequence, but it is worth mentioning that syntenic ORFs of analogous size and positions have been detected in other phasmavirids [18].

A phylogenetic tree based on LbPV RdRp and viruses belonging to different families within the class *Bunyaviricetes* showed that LbPV is clustered with members belonging to the *Phasmaviridae* family, order *Elliovirales* (Figure 1C). A Circos plot of spatial identity clearly supported the best complete coverage of LbPV only with phasmaviruses (Figure 1F). A phylogenetic tree based on LbPV RdRp and viruses belonging to members of different genera within the *Phasmaviridae* family showed that this virus is grouped with those viruses belonging to the *Orthophasmavirus* genus in a clade with the recently described moth-associated PBV2 and SEPV (Figure 1G), showing a close evolutionary relationship of the moth-associated viruses within the *Orthophasmavirus* genus.

#### 3.2.2. A Novel Carmotetravirus Linked to *L. botrana*

A novel carmotetravirus was identified and named *Lobesia botrana* carmotetravirus (LbCaV) (GenBank accession number BK067727). This virus was detected in ten transcriptome datasets corresponding to larval samples hosted by two different *V. vinifera* cultivars exposed to two different CO_2_ levels during two developmental stages (Table 1). The positive-sense, single-stranded RNA genome of LbCaV is composed of 6575 nucleotides (nt) and has three main ORFs (Figure 2A). One ORF encodes the replicase fusion protein (p123) of 1087 aa. This ORF contains a readthrough stop codon, which results in the translation of a putative p58 protein of 521 aa or the fusion protein p123 (Figure 2A). The p123 shows the highest BlastP similarity with the RdRp protein encoded by Hangzhou sesamia inferens carmotetravirus 1 (HSICTV1) with a 45.97% identity (Appendix A) and includes a RDRP_SSRNA_POS conserved domain (Carmo_RDRP4, 8FMA_O RdRP, E-value 2.5 × 10^−18^) in its sequence at aa positions 737–908 (Figure 2A). The p58 protein shows the highest BlastP similarity with a hypothetical protein encoded by HSICTV1 with a 24.71% identity, and no conserved domains were identified in its sequence, but two transmembrane regions were (Figure 2A). Another ORF, upstream of and overlapping with the replicase, encodes a putative protein p130 of 1201 aa (Figure 2A). This protein shows the highest BlastP similarity with a hypothetical protein encoded by HSICTV1 with a 27.49% identity (Appendix A), and no conserved domain was identified in its sequence. The third main ORF, which is located downstream of that one encoding the replicase, encodes a putative coat protein (p87) of 796 aa (Figure 2A). The CP shows the highest BlastP similarity with the CP protein encoded by the Providence virus (PrV), the member of the only International Committee on Taxonomy of Viruses (ICTV)-recognized species within the single genus *Alphacarmotetravirus* of the family *Carmotetraviridae*, with a 50.51% identity (Appendix A), and the Viral_coat conserved domain was identified in its sequence at aa positions 271–443 and 508–695 (Figure 2A). Phylogenetic analysis based on the LbCaV replicase clustered this virus with a clade including the moth-associated PrV and other putative members of the *Carmotetraviridae* family described on metagenomic studies and away from the clades represented with members of the related *Permutetraviridae* and *Alphatetraviridae* families (Figure 2B).

#### 3.2.3. A Novel Cypovirus Linked to *L. botrana*

A novel cypovirus was identified and named *Lobesia botrana* cypovirus (LbCPV). This cypovirus was detected in 24 transcriptome datasets corresponding to larval samples hosted by two different *V. vinifera* cultivars exposed to two different CO_2_ levels during two developmental stages (Table 1). The LbCPV genome is composed of ten double-stranded RNA (dsRNA) segments (Figure 3A) (GenBank accession numbers BK067728-BK067737). The lengths of the ten segments range from 4070 nt to 853 nt. Segment 1’s length is 4070 nt, and it has one ORF that encodes a putative protein of 1331 aa. This protein shows the highest BlastP similarity with the Clanis bilineata cypovirus-type 23 (CbCPV-23) VP1 with an 88.43% identity (Appendix A), and a Capsid protein VP1 3JB0_B conserved domain was identified in its sequence at aa positions 151–1331. Therefore, this protein is the putative major capsid protein (Figure 3A). Segment 2’s length is 3710 nt, and it has one ORF that encodes a putative protein of 1183 aa. This protein shows the highest BlastP similarity with the Daphnis nerii cypovirus-type 23 (DnCPV-23) VP2 with an 87.15% identity (Appendix A), and the CPV_RdRp_N, CPV_RdRp_pol_dom, and CPV-RdRp_C Cypovirus 6K32_A conserved domains were identified in its sequence at aa positions 48–296, 327–697, and 847–1179, respectively. Therefore, this protein is the putative RdRp protein (Figure 3A). Segment 3’s length is 3308 nt, and it has one ORF that encodes a putative protein of 1071 aa. This protein shows the highest BlastP similarity with the DnCPV-23 VP3 with an 84.59% identity (Appendix A), and the 3JB0_A motif (9–1069, E-value 2.7 × 10^−120^) was detected including a Reov_VP3_GTase, Reov_VP3_MTase1, and Reov_VP3_MTase2 conserved domains in its sequence at aa positions 11–277, 483–703, and 872–1067, respectively. Therefore, this protein is a putative methyltransferase and guanylyltransferase structural protein (Figure 3A). Segment 4’s length is 3781 nt, and it has one ORF that encodes a putative protein of 1245 aa. This protein shows the highest BlastP similarity with the CbCPV-23 VP4 with a 79.87% identity (Appendix A), and a PPPDE domain-containing protein; Cell attachment, Membrane penetration, and VIRAL PROTEIN of Bombyx mori cypovirus 1 (7WHM_A, E-value 2 × 10^−102^) was identified at aa positions 19–1229. Thus, this protein is a putative minor capsid protein. Segment 5’s length is 2049 nt, and it has one ORF that encodes a putative protein of 642 aa. This protein shows the highest BlastP similarity with the DnCPV-23 VP5 with a 56.44% identity (Appendix A). Its DnCPV-23 counterpart was suggested to be a protein with an unknown function [19]. Interestingly, the recently described protein domain named “widespread, intriguing, versatile” (WIV) was detected in several cypoviruses, including DnCPV-23 [20]. Segment 6’s length is 2000 nt, and it has one ORF that encodes a putative protein of 612 aa. This protein shows the highest BlastP similarity with the DnCPV-23 VP6 with a 78.43% identity (Appendix A) and no conserved domains. Segment 7’s length is 1833 nt, and it has one ORF that encodes a putative protein of 537 aa. This protein shows the highest BlastP similarity with the CbCPV-23 VP7 with an 81.94% identity (Appendix A), and a Viral structural protein 4 (6K32_B, E-value 1.1 × 10^−42^); Cypovirus, VIRAL PROTEIN-RNA complex; and HET: A2M conserved motif was identified at aa positions 13–530. Thus, this protein is a putative structural protein. Segment 8’s length is 1247 nt, and it has one ORF that encodes a putative protein of 391 aa. This protein shows the highest BlastP similarity with the DnCPV-23 VP8 with an 83.33% identity (Appendix A), and a VP5 conserved domain (31Z3_D, E-value 8.7 × 10^−59^) was identified in its sequence at aa positions 3–282. Thus, this protein is a putative structural protein. Segment 9’s length is 1105 nt, and it has one ORF that encodes a putative protein of 300 aa. This protein shows the highest BlastP similarity with the DnCPV-23 VP9 with a 79.26% identity (Appendix A), and an NP protein, nucleocapsid protein, nucleoprotein, VIRAL PROTEIN motif was identified at aa positions 107–189 (4J4Y_D). Segment 10’s length is 853 nt, and it has one ORF that encodes a putative protein of 246 aa. This protein shows the highest BlastP similarity with the DnCPV-23 VP10 with a 97.56% identity (Appendix A), but the nt identity between LbCPV, DnCPV, and CbCPV in segment 10 is ca. 80%, which is in the range of the species demarcation criteria based on this segment for cypoviruses by the ICTV. Moreover, as expected, a CPV_Polyhedrin conserved domain (5A99_A, E-value 5.5 × 10^−53^) was identified in its sequence at aa positions 3–244. Therefore, this protein is a putative polyhedrin protein (Figure 3A).

Phylogenetic analysis based on the LbCPV replicase placed this virus within the Lepidoptera, mostly moths, associated *Cypovirus* genus, clustering most closely with the recently described DnCPV-23 and CbCPV-23, suggesting the possibility that these three viruses could be members of a new species within genus *Cypovirus* (Figure 3B).

#### 3.2.4. A Novel Bi-Segmented Sobemo-like Virus

A novel sobemo-like virus was identified and named *Lobesia botrana* sobemo-like virus (LbSV). This virus was detected in 27 transcriptome datasets, including larval samples hosted by two different V. vinifera cultivars exposed to two different CO_2_ levels during two developmental stages and whole adults susceptible or resistant to insecticides sampled in both Germany and Turkey (Table 1). The sequences of two strains, LbSV_Ger (GenBank accession numbers BK067738-BK067739) and LbSV_Tur (GenBank accession numbers BK067740-BK067741), were assembled in this study. The LbSV_Ger and LbSV_Tur genomes comprised two segments of single-stranded, positive-sense RNA (Figure 4A). The RNA1 is composed of 2701 nt and 2622, respectively, while RNA 2 comprises 1413 and 1379 nt, respectively (Figure 4). LbSV_Ger and LbSV_Tur segment 1 have two ORFs named HP CDS and RdRP CDS. ORF HP encodes a hypothetical protein (HP), and an RdRP is encoded by the other and is expressed as a fusion polyprotein through a −1 ribosomal frameshift mechanism (Figure 4A). The -1RFM signal consists of the slippery sequence 5′-GGGAAGC-3′ at coordinates 1221–1227. LbSV_Ger and LbSV_Tur HP have 459 aa and are 98% identical. This protein shows the highest BlastP similarity with Latepeofons virus (LtPV), with a 49.14% identity and 48.15% identity, respectively. A Peptidase_S1_PA conserved domain was identified in their sequence at aa positions 97–285 (Pro-VPg, 6FF0_A, Figure 4A). LbSV_Ger and LbSV_Tur RdRp have 865 aa and are 98% identical. This protein shows the highest BlastP similarity with Wugcerasp virus 3 (WV3), with a 53.96% identity and 54.20% identity, respectively. An RNA-dir_pol-C was identified in their sequence (6QWT_A) at aa positions 510–851. LbSV_Ger and LbSV_Tur segment 2 have one ORF that encodes the CP with a size of 416 aa (Figure 4A). Their CP are 98.3% identical and show the highest BlastP similarity with Buhirugu virus 16 (BHRGV16) CP protein with a 37.97% identity and 37.88% identity, respectively. A Nodavirus_capsid conserved domain (4WIZ_CX, E-value 2.9 × 10^−21^) was identified in their sequence at aa positions 76–361 (Figure 4A). Phylogenetic analysis based on the LbSV_Ger placed this virus within the clade containing unclassified solemovirids identified from a variety of arthropods, clustering most closely with the wasp-associated WV3 (Figure 4B).

#### 3.2.5. A Glimpse into Expression Level Profiles and Preliminary Prevalence of *L. botrana* Viruses

We investigated the presence and absence of specific RNA viruses in *Lobesia botrana* moths from Germany and Turkey. We analyzed both adults and larvae parasitizing Cabernet Sauvignon or Riesling grapevines at different growth stages under conditions of high and low CO_2_ levels.

Initial analyses revealed varying expression level profiles and prevalence of these viruses across different samples. The RNA expression levels of the four viruses associated with *L. botrana*—LbPV, LbSV, LbCPV, and LbCaV—were evaluated across the 29 RNA libraries, as shown in Figure 5A. The relative abundance of viral RNA, measured as fragments per kilobase of transcript per million mapped reads (FPKM), demonstrated marked variability across libraries. LbCPV was the most consistently and abundantly expressed virus across the libraries, followed by LbSV, LbCaV, and LbPV. Notably, certain libraries exhibited minimal or undetectable levels of viral RNA, reflecting a potential library-specific variability in viral presence or absence. Figure 5B illustrates the RNA relative expression levels of the 10 genome segments of LbCPV. Each segment displayed distinct expression profiles, with segment 10—encoding the polyhedrin protein—showing the highest expression levels. This pattern aligns with the functional importance of polyhedrin in viral particle assembly [21]. While the expression of other segments was relatively uniform, minor variability among libraries suggests differential segment replication or transcript stability.

Figure 5C–E presents the RNA levels of LbCPV, LbSV, and LbCaV under varying environmental conditions, grapevine cultivars, and developmental stages of *V. vinifera*. Under elevated CO_2_ conditions, LbCPV exhibited slightly higher FPKM values compared to ambient CO_2_, suggesting that elevated CO_2_ may enhance viral RNA expression (Figure 5C). In contrast, LbSV (Figure 5D) and LbCaV (Figure 5E) showed no significant differences in expression levels between the two CO_2_ conditions, suggesting the possibility that these viruses may be less responsive to atmospheric CO_2_ variations. Viral RNA levels were also compared between two grapevine cultivars: Cabernet Sauvignon and Riesling. For LbCPV, FPKM values were slightly higher in Riesling than in Cabernet Sauvignon, suggesting a potential host effect for this virus. Similarly, LbSV displayed marginally higher expression in Riesling, while LbCaV exhibited negligible differences between the two cultivars. The RNA expression of LbCPV varied between flowering and veraison stages, with higher levels observed during veraison (Figure 5C). A similar trend was noted for LbSV (Figure 5D). For LbCaV (Figure 5E), RNA levels were lower overall, with only a minor increase during veraison, suggesting either mild or no effect of the cv. host on RNA virus titters. Nevertheless, caution should be exercised when interpreting these RNA virus expression levels due to the relatively low sample size, which may limit the generalizability of the findings. Additionally, other unassessed variables, such as environmental stressors, microbial interactions, or host physiological conditions, may influence viral RNA levels and confound the observed patterns. Further investigations are warranted to elucidate the ecological and epidemiological factors driving the observed patterns and to assess the potential implications on the host.

## 4. Discussion

### 4.1. A Glimpse into L. botrana Viromics

Our high-throughput sequencing (HTS)-based exploration of the *L. botrana* virome reveals a plethora of novel viral diversity. The discovered cypovirus lineage within *L. botrana* aligns with growing evidence highlighting host specialization and adaptation within Lepidoptera-associated viruses [22]. Notably, our analysis suggests potential inter-species transmission dynamics within *L. botrana* populations, echoing recent studies indicating horizontal virus transfer as a driving force in shaping insect viral communities and evolution [23,24]. Further investigation into this cypovirus lineage, employing phylogenetic and comparative genomics approaches, will provide valuable insights into its evolutionary history and ecological roles within *L. botrana* populations.

As expected, LbCPV was the most consistently and abundantly expressed virus across all the libraries analyzed. It is well known that, in general, cypoviruses infect midgut epithelial cells of insect larvae and are able to replicate in huge amounts, causing chronic rather than lethal diseases in infected insect hosts [25]. Therefore, among the Lobesia-associated viruses identified in our study, LbCPV could be a promising agent to manage *L. botrana* populations.

### 4.2. A Novel Phasmavirus

*Phasmaviridae* is a recently described family of negative-strand RNA viruses that are maintained in and/or transmitted by insects [26]. The LbPV genome resembles the one reported for phasmavirids, which possess a segmented genome of three segments (L, M, and S) that encode RNA-dependent RNA polymerase, a glycoprotein precursor, and the nucleocapsid protein [26]. LbPV-encoded RdRps have all the motifs identified in the bunyavirales-encoded RdRps, which are essential for genome replication and transcription [27]. As was reported for the G precursor protein encoded by phasmavirids [26], the LbPV G precursor is processed by a conserved signal peptide peptidase to yield the mature Gn and Gc proteins, which mediate cell entry of phasmavirions [28]. The LbPV S segment encodes a putative NSs protein upstream of the NP, which was also found in some phasmavirids [26,28], including the moth-associated PBV2 [29]. Moreover, the 3′ terminal nucleotides are conserved among the three LbPV segments and are likely complementary to the terminal sequences found in the 5′ end, which support the formation of panhandle-like structures that play a key role in RNA synthesis and genome packaging [30]. Conserved terminal segmentation among all three segments is characteristic of phasmavirids, and it was suggested that the first seven of the terminal nucleotides may be family-specific [31].

LbPV represents the third reported phasmavirid identified in Lepidoptera, as this family was previously known to infect mostly mosquitoes and midges [26]. The three moth-associated phasmavirids clustered together in the phylogenetic tree indicate a common evolutionary trajectory of these viruses, suggesting a probable host–virus co-evolution. The BlastP searches, as well as the phylogenetic analysis, confirmed the placement of LbPV within the *Orthophasmavirus* genus. This virus should be classified as a novel member belonging to the *Orthophasmavirus* genus since the aa sequence identity of the L protein between the LBPV and related viruses is below 95%, which is the species demarcation criteria for this genus [26]. Unraveling the evolutionary trajectory and transmission dynamics of this novel orthophasmavirus within *L. botrana* demands further research, potentially elucidating novel ecological interactions within arthropod communities.

### 4.3. A Novel Carmotetravirus

The novel carmotetravirus identified in this study has a genome that exhibits the typical genomic organization of carmotetraviruses, which encodes three main ORFs instead of the two ORFs that other tetraviruses have [32]. LbCaV, like the carmotetravirus PrV, has a replicase with a readthrough stop signal, resulting in the production of an accessory protein. LbCaV replicase and accessory protein molecular weights are higher than that reported for PrV [32]. The largest ORF, which overlaps the replicase, has a protein with a similar size in LbCTV and PvR, and the presence, but with a low E-value, of a putative 2A-like processing sequence, like the one located at the N terminus of PrV p130 [32]. Carmotetraviruses, belonging to the family *Carmotetraviridae*, possess a single-stranded, positive-sense RNA genome; only one genus, the *Alphacarmotetravirus*, has been created within this family, and its sole member, PrV, has lepidopteran as its natural host [33]. Phylogenetic relationships grouped LbCTaV with PrV and some lepidopteran-associated proposed and recently described carmotetraviruses. Thus, LbCaV could be classified as a novel member of a species within the *Alphacarmotetravirus* genus. The evolutionary history of PrV is not well understood; the expression of PrV is similar to that observed in plant-infecting Tombusviruses, and PrV non-structural proteins are structurally related to plant tombusvirus and umbravirus accessory proteins [34]. Ritah proposed that PrV is a hybrid virus with the potential to infect and replicate in both host plant and animal systems [34]. This hypothesis was later confirmed by [35], who presented evidence that PrV can establish a productive infection in plants as well as in animal cells. Therefore, further studies should further characterize the biology and evolution of LbCaV.

### 4.4. A Novel Cypovirus

The novel cypovirus identified in this study has a genome that exhibits the typical segmented genomic organization of cypoviruses [36]. Cypoviruses, belonging to the family *Spinareoviridae*, represent a diverse and widely distributed group of segmented double-stranded RNA (dsRNA) viruses infecting a variety of arthropod hosts. So far, 16 different types of cypoviruses have been recognized by the ICTV, but many more cypoviruses are still to be recognized (https://ictv.global/report/chapter/spinareoviridae/spinareoviridae/cypovirus) (accessed on 15 November 2024). Their segmented genome structure, a hallmark of the Reovirales order, offers unique features and flexibility in their evolutionary potential [36]. Each cypovirus genome typically consists of 10 dsRNA segments, each encoding one or two proteins crucial for viral replication, assembly, and pathogenesis. This modularity allows for intra- and inter-species reassortment, potentially contributing to the emergence of new viral strains and adaptation to diverse hosts [36]. Structural and functional annotation based on conserved domains identified features typical of cypoviruses in those LbCPV-encoded proteins, including RdRp, capsid proteins, and virion assembly proteins, which are essential components for viral replication, packaging, and assembly [37,38,39,40,41]. Moreover, the recently described domain WIV, which is a potential virulence factor that facilitates the infection of arthropods [20], was also identified in the LbCPV-encoded P5 protein. LbCPV is closely related to CbCPv-23 and DnCPV-23; however, the possibility of being recognized as a distinct species rather than as a novel isolate of the cypovirus 23 type since the nt identity for segment 10 is around the value set by the ICTV to demarcate species within the genus *Cypovirus* (https://ictv.global/report/chapter/spinareoviridae/spinareoviridae/cypovirus) (accessed on 15 November 2024) should be explored. DnCPV-23 was identified in the oleander hawk moth [16], while CbCPV-23 was identified in the soybean hawk moth [42]. Cypoviruses have been suggested as promising agents for the control of lepidopteran species [43]. Interestingly, the closely related DnCPV-23 has a lethal effect on its host [19,44]. Thus, LbCPV could be a promising virus for the control of *L. botrana*; therefore, further studies focused on the LbCPV effect on its host should be carried out.

### 4.5. A Novel Sobemo-like Virus

*Solemoviridae* is a recently established family of positive-strand RNA viruses whose recognized members infect plants [45]. Typically, solemovirid genomes are unsegmented with four to ten ORFs, which code for a viral suppressor of RNA silencing, a polyprotein (which contains a serine protease and an RdRp conserved domains), and a CP [45]. However, recently, sobemo-like viruses with two segments were identified in arthropods such as crustaceans [46], myriapods [47], and insects [47,48,49,50]. Here, we identified the sobemo-like virus LbSV, which adds to the group of sobemo-like viruses with bi-segmented genomes infecting non-plant hosts and is likely the first sobemo-like virus identified in moths. The genomic organization of segment 1 of LbSV and those bi-segmented sobemo-like viruses is highly similar to that one reported for the 5′ terminal half region of the solemovirids (the ORF encoding the viral suppressor of RNA silencing protein is not present), because the first ORF encodes a polyprotein with a serine protease domain and the RdRp is translated via −1 programmed ribosomal frameshift (−1 PRF) from the next ORF [45]. LbSV’s slippery sequence 5′-GGGAAGC-3′ is highly similar to the one reported for poleroviruses, which is 5′-GGGAAAC-3′ [45]. Interestingly, LbSV CP, as was reported for other sobemo-like viruses CP [46], not only shares sequence similarity with other sobemo-like viruses but also with noda-like viruses and permutotetra-like viruses. It has been hypothesized that because of capsid genes being encoded by subgenomic RNAs or genomic segments in these viruses, there is a possibility that horizontal gene transfer could have been the outcome of mispackaging during co-infection of the same host [46]. Phylogenetic analysis based on the RdRp sequences placed LbSV within the clade containing unclassified solemovirids. Thus, based on genomic organization and phylogenetic relationships, at least a novel genus within the family *Solemoviridae* should be established to classify those sobemo-like viruses that have arthropods as their hosts.

### 4.6. An Emergent Landscape of the L. botrana Virome

Beyond individual viral characterization, our analysis highlights the potential influence of the *L. botrana* virome on the moth–grapevine ecosystem. The genomic architecture of the novel viruses identified in this study provides valuable clues about their replicative strategies and evolutionary relationships. The segmented dsRNA genome of the cypovirus and the monosegmented carmotetravirus align in genomic organization with their established family counterparts, suggesting potential shared evolutionary pathways within these viral lineages. On the other hand, the genomic arrangement of the sobemo-like virus represents atypical features within its respective family, prompting further investigation into its replicative mechanisms and evolutionary origins.

Moving forward, our research opens exciting avenues for exploring the functional roles of these newly discovered viruses within *L. botrana*. Employing virus–host interaction studies and functional assays will unlock crucial insights into how these viruses influence their host’s physiology, behavior, and susceptibility to environmental factors. Understanding these interactions could pave the way for the development of novel pest-control strategies, potentially targeting specific viral processes or manipulating virus–host interactions to interfere with pest biology. The unveiled diversity of the *L. botrana* virome also calls for continued exploration of this hidden viral landscape. Advanced sequencing approaches, such as metagenomics and metatranscriptomics, hold immense potential for uncovering even greater viral diversity within *L. botrana* and other Lepidoptera species. Furthermore, functional screening techniques aimed at identifying virus-encoded virulence factors or host immunity pathways could offer valuable tools for elucidating the intricate interplay between *L. botrana* and its virome.

Nevertheless, we used raw data obtained by other researchers and we do not know if the larvae used for RNA extraction were infected by acute viral infection (with morphological symptoms) or chronic, without symptoms. Since some of the revealed viruses/strains may form chronic infections (for example, CPVs), the result of our study is restricted only by the demonstration of the presence of potentially entomopathogenic viruses in *Lobesia botrana*. For a more convincing recommendation to consider CPV as a potential insecticide, additional studies (which are behind the main idea of the current study) are needed.

### 4.7. Future Perspectives

While our initial investigation has unveiled the diversity of novel viruses residing within the European grapevine moth, a more profound understanding of their influence within this agricultural pest and the vineyard ecosystem necessitates further exploration. This necessitates deeper insight into the intricate relationship between virus and host, elucidating the specific roles these viruses play in shaping *L. botrana*’s biology and ecological interactions. Employing virus–host interaction studies using molecular and physiological techniques will allow us to observe the dynamic interplay between *L. botrana* and its virome, providing insights into how these viruses influence the moth’s metabolism, behavior, and resilience against environmental stressors.

The vineyard ecosystem is a complex web of interdependencies, and our newfound understanding of the *L. botrana* virome offers a glimpse into its hidden holobiome. This knowledge could pave the way for innovative biocontrol solutions that target vectors, disrupting the spread of plant diseases before they take root.

## 5. Conclusions

Our high-throughput sequencing investigation has fostered our understanding of the European grapevine moth by unveiling a previously unseen viral world residing within this agricultural pest. The discovery of four novel viruses belonging to diverse families reflects a complex underexplored viral community potentially influencing *L. botrana* biology and impacting the broader vineyard ecosystem.

The broad presence of these viruses in public metatranscriptomic datasets suggests their widespread prevalence and potential ecological significance. Additionally, the genomic architecture of each novel virus provides valuable clues about their replicative strategies and evolutionary relationships, opening doors for further exploration.

This research stands as a pivotal step in comprehending the *L. botrana* virome and its ecological implications. Unveiling this hidden dimension within *L. botrana* biology opens exciting avenues for sustainable pest management and a deeper understanding of the intricate viral webs shaping agricultural ecosystems. By embracing future research directions, we can harness the vast potential hidden within the *L. botrana* virome, ensuring a more sustainable future for vineyards and a broader understanding of the moth holobiome.

## Figures and Tables

**Figure 1 viruses-17-00095-f001:**
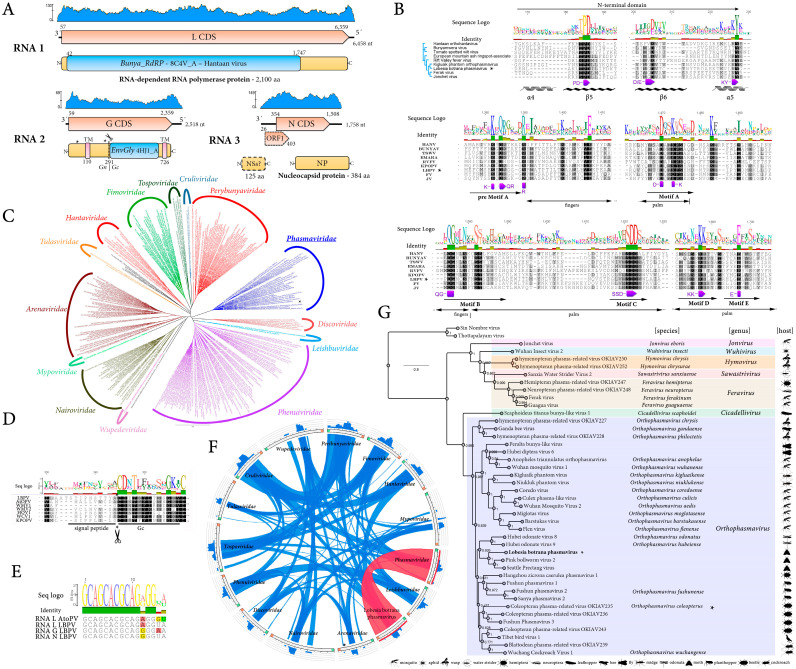
Genome organization, sequence conservation, phylogenetic relationships, and evolutionary insights into *Lobesia botrana* phasmavirus (LbPV) and related viruses. (**A**) Schematic representation of the genome organization of LbPV, showing its three-segmented RNA genome. RNA 1 encodes the RNA-dependent RNA polymerase (RdRP), RNA 2 encodes the glycoprotein (G), and RNA 3 encodes the nucleocapsid protein (N). The annotations highlight functional domains, conserved motifs, and transmembrane regions (TM). Coding sequences are depicted in orange arrowed rectangles, predicted proteins in yellow rectangles, and conserved domains in blue rectangles. The coverage plot (blue) displays the read depth across the genome, and the number represents the maximum coverage. (**B**) Sequence conservation and alignment of the RdRP of LbPV and other bunyavirids. Sequence logos indicate conserved residues across related viruses. Functional motifs (e.g., Motifs A, B, C, D, E) are labeled, and structural elements (e.g., α-helices and β-sheets) are indicated. (**C**) Circular phylogenetic tree showing relationships among members of *Phasmaviridae* and other viral families within the *Bunyavirales* order. Clades corresponding to different families are color-coded, and the tree highlights the evolutionary placement of LbPV within *Phasmaviridae*. (**D**) Sequence alignment of the glycoprotein cleavage site, showing conservation of the signal peptide and glycosylation motifs among LbPV and *Phasmaviridae* members. (**E**) Sequence logo of conserved terminal sequences of the 3′ RNA segments indicating base-pairing complementarity. (**F**) Circos plot depicting shared spatial protein domain architectures and sequence similarities across viral families within *Bunyavirales*. Connections are highlighted in blue (low similarity) to red (high similarity). (**G**) Maximum-likelihood phylogenetic tree of representative *Phasmaviridae* species, including members of genus *Orphophasmavirus* and closely related genera. The tree includes FastTree support values and host associations, with icons representing host types. Virus members of ICTV-recognized species are indicated with binomial nomenclature. The scale bar represents the number of amino acid substitutions per site.

**Figure 2 viruses-17-00095-f002:**
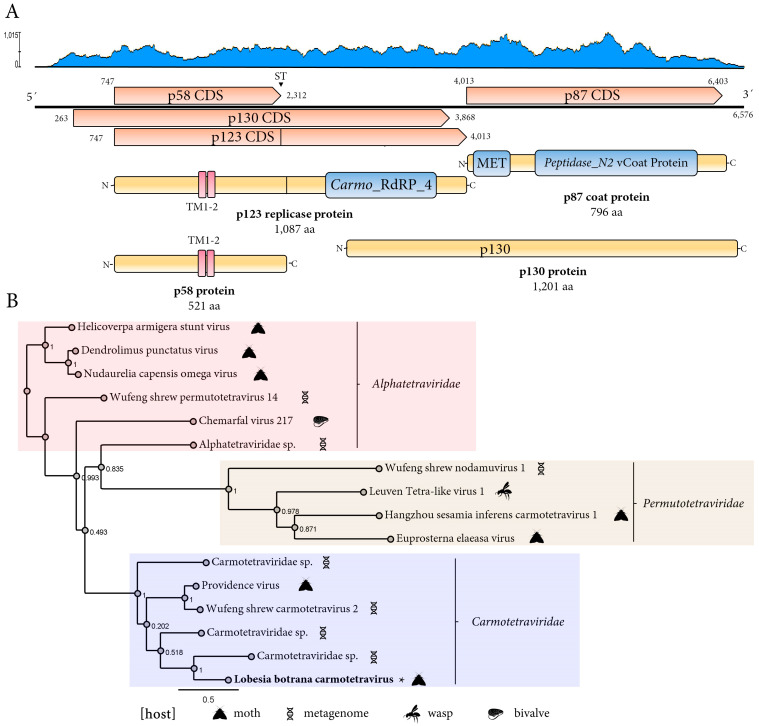
Genome organization and phylogenetic relationships of *Lobesia botrana* carmotetravirus (LbCaV) and related viruses. (**A**) Schematic representation of the genome organization of LbCaV. The linear genome is annotated with key coding sequences (CDS), including the replicase proteins (p123), the p130, and the coat protein (p87). Functional domains are highlighted, including Carmo_RdRP_4 (RNA-dependent RNA polymerase), MET (methyltransferase), and Peptidase_N2. Transmembrane regions (TM1-2) are indicated in the p58 protein, while protein sizes in amino acids (aa) are shown for major proteins. The coverage plot (blue) displays the read depth across the genome, with peaks corresponding to coding regions. Coding sequences are depicted in orange arrowed rectangles, predicted proteins in yellow rectangles, and conserved domains in blue rectangles. (**B**) Maximum-likelihood phylogenetic tree of viruses from *Carmotetraviridae*, *Alphatetraviridae*, and *Permutotetraviridae*. Clades corresponding to each family are shaded (*Carmotetraviridae* in blue, *Alphatetraviridae* in pink, and *Permutotetraviridae* in beige). Host associations are indicated with icons. FastTree support values are shown at nodes, and representative species are labeled. The scale bar represents the number of amino acid substitutions per site. This tree highlights the evolutionary relationships and host diversity among these virus families.

**Figure 3 viruses-17-00095-f003:**
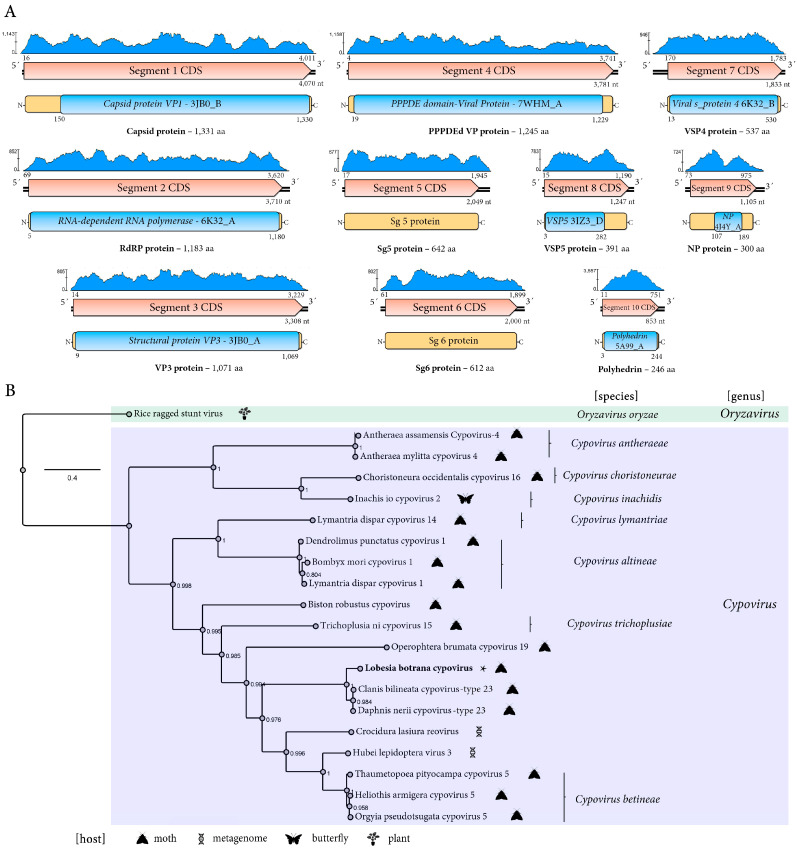
Genomic organization and phylogenetic analysis of *Lobesia botrana* cypovirus (LbCPV). (**A**) Genomic organization of LbCPV. The genome is segmented into 10 double-stranded RNA (dsRNA) segments (Seg 1–10). Each segment contains a single open reading frame (ORF) encoding a single protein. The predicted functions of the encoded proteins are indicated. The coverage plot (blue) displays the read depth across the genome. Coding sequences are depicted in orange arrowed rectangles, predicted proteins in yellow rectangles, and conserved domains in blue rectangles. (**B**) Phylogenetic tree of LbCPV and cypoviruses based on the amino acid sequences of the replicase protein. The tree used is an outgroup of the orizavirus Rice ragged stunt virus. FastTree support values are shown at nodes, and representative species are labeled. The scale bar represents the number of amino acid substitutions per site. The hosts of the viruses are indicated by icons.

**Figure 4 viruses-17-00095-f004:**
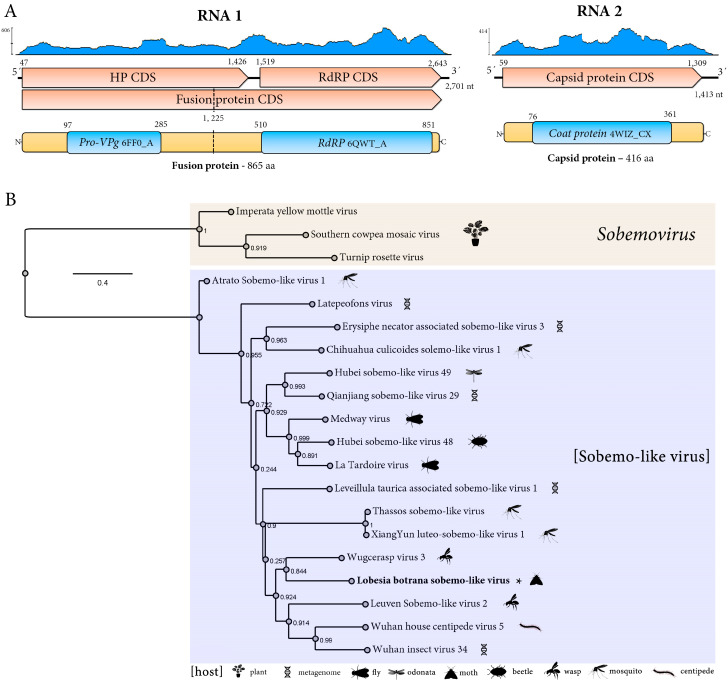
Genomic organization and phylogenetic analysis of *Lobesia botrana* sobemo-like virus (LbSV). (**A**) Genomic organization of LbSV. The genome is segmented into two positive single-strand RNA (+ssRNA) segments (RNA 1 and RNA 2). The predicted functions of the encoded proteins are indicated. The coverage plot (blue) displays the read depth across the genome. Coding sequences are depicted in orange arrowed rectangles, predicted proteins in yellow rectangles, and conserved domains in blue rectangles. (**B**) Phylogenetic tree of LbSV, sobemovirus, and sobemo-like viruses based on the amino acid sequences of the replicase protein. FastTree support values are shown at nodes, and representative species are labeled. The scale bar represents the number of amino acid substitutions per site. The hosts of the viruses are indicated by icons.

**Figure 5 viruses-17-00095-f005:**
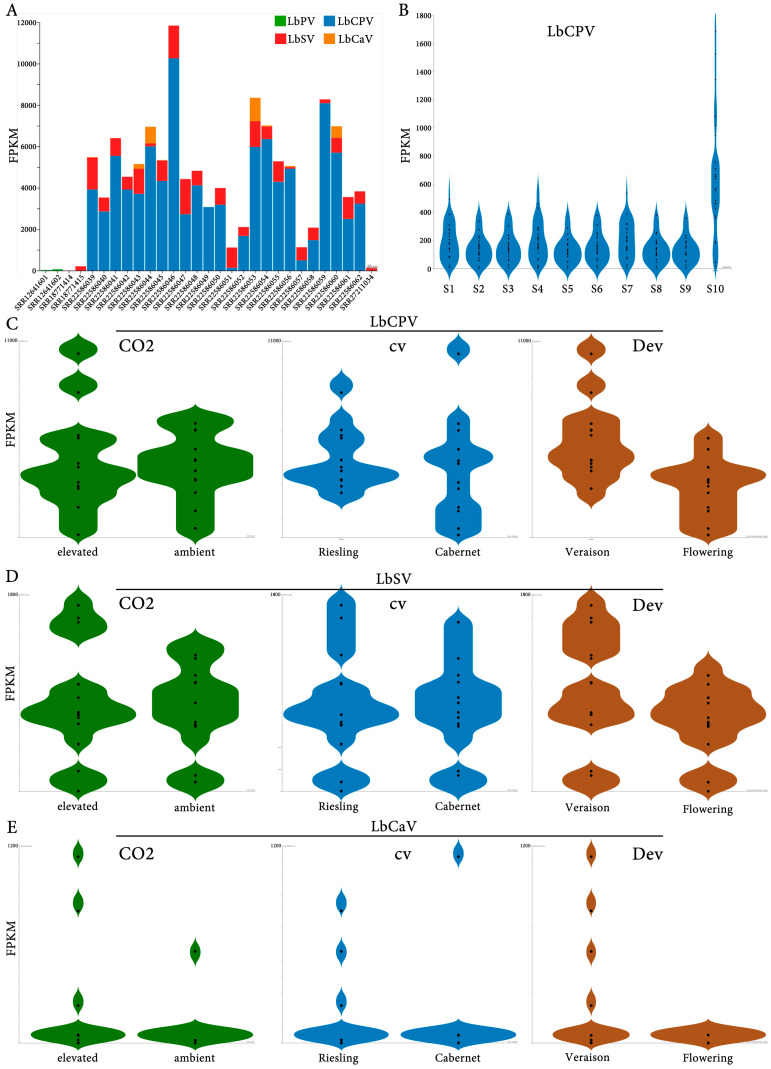
Assessment of relative RNA levels landscape of four *Lobesia botrana*-associated viruses in 29 RNA-seq sequencing libraries from various experimental conditions. (**A**) FPKM (Fragments Per Kilobase of transcript per million mapped reads) values for four viruses associated with the grapevine moth, *L. botrana*: LbPV (*L. botrana* phasmavirus, green), LbSV (*L. botrana* sobemo-like virus, red), LbCPV (*L. botrana* cypovirus, blue), and LbCaV (*L. botrana* carmotetravirus, orange). (**B**) FPKM values for each of the 10 genome segments of LbCPV. Segment 10 encodes the polyhedrin protein, a structural component of the virus capsid. (**C**–**E**) FPKM values for LbCPV (**C**), LbSV (**D**), and LbCaV (**E**) in *Vitis vinifera* plants subjected to different environmental and developmental conditions: CO_2_: plants were exposed to elevated (480 ppm) or ambient (400 ppm) CO_2_ levels. cv: moths were bred on two different grapevine cultivars: Cabernet Sauvignon and Riesling. Dev: moths were bred at two distinct developmental stages: flowering and veraison (onset of berry ripening). Each violin plot represents the distribution of FPKM values across multiple biological replicates. The width of the violin plot is proportional to the density of data points at that FPKM value.

**Table 1 viruses-17-00095-t001:** *Lobesia botrana* high-throughput RNA sequencing library datasets assessed for virus discovery. The runs are publicly available at the NCBI-SRA archive. Abbreviations: virusRPM, total virus reads per million reads in the specific run; LbCPV, *Lobesia botrana* cypovirus RPM; LbSV, *Lobesia botrana* sobemo-like virus RPM; LbPV, *Lobesia botrana* phasmavirus RPM; LbCaV, *Lobesia botrana* carmotetravirus RPM.

Run	BioProject	BioSample	Bases	SRA Study	virusRPM	LbCPV	LbSV	LbPV	LbCaV
SRR12641601	PRJNA663283	SAMN16125021	11.76 G	SRP282412	26	0	0	26	0
SRR12641602	PRJNA663283	SAMN16125021	11.22 G	SRP282412	62	0	0	62	0
SRR18771414	PRJNA827155	SAMN27603370	7.76 G	SRP370512	8	0	8	0	0
SRR18771415	PRJNA827155	SAMN27603369	10.09 G	SRP370512	211	0	211	0	0
SRR22586039	PRJNA910346	SAMN32123568	3.52 G	SRP412145	5486	3925	1544	0	17
SRR22586040	PRJNA910346	SAMN32123567	3.19 G	SRP412145	3533	2859	674	0	0
SRR22586041	PRJNA910346	SAMN32123566	3.58 G	SRP412145	6403	5547	856	0	0
SRR22586042	PRJNA910346	SAMN32123565	3.73 G	SRP412145	4537	3920	617	0	0
SRR22586043	PRJNA910346	SAMN32123564	3.23 G	SRP412145	5152	3714	1213	0	226
SRR22586044	PRJNA910346	SAMN32123563	3.71 G	SRP412145	6954	6008	145	0	801
SRR22586045	PRJNA910346	SAMN32123562	4.16 G	SRP412145	5347	4331	997	0	18
SRR22586046	PRJNA910346	SAMN32123582	3.78 G	SRP412145	11,846	10,261	1583	0	1
SRR22586047	PRJNA910346	SAMN32123581	4.17 G	SRP412145	4428	2728	1699	0	1
SRR22586048	PRJNA910346	SAMN32123580	3.85 G	SRP412145	4825	4125	700	0	0
SRR22586049	PRJNA910346	SAMN32123579	3.27 G	SRP412145	3072	3071	1	0	0
SRR22586050	PRJNA910346	SAMN32123561	3.57 G	SRP412145	3995	3187	808	0	0
SRR22586051	PRJNA910346	SAMN32123578	3.08 G	SRP412145	1117	138	979	0	0
SRR22586052	PRJNA910346	SAMN32123577	3.89 G	SRP412145	2108	1677	431	0	0
SRR22586053	PRJNA910346	SAMN32123576	3.68 G	SRP412145	8353	5980	1243	0	1131
SRR22586054	PRJNA910346	SAMN32123575	4.14 G	SRP412145	7016	6360	610	0	46
SRR22586055	PRJNA910346	SAMN32123574	4.49 G	SRP412145	5283	4293	990	0	0
SRR22586056	PRJNA910346	SAMN32123573	3.48 G	SRP412145	5058	4926	84	0	48
SRR22586057	PRJNA910346	SAMN32123572	5.20 G	SRP412145	1131	498	633	0	0
SRR22586058	PRJNA910346	SAMN32123571	3.34 G	SRP412145	2077	1474	603	0	0
SRR22586059	PRJNA910346	SAMN32123570	3.48 G	SRP412145	8370	8093	277	0	0
SRR22586060	PRJNA910346	SAMN32123569	3.96 G	SRP412145	6978	5703	721	0	554
SRR22586061	PRJNA910346	SAMN32123560	4.31 G	SRP412145	3555	2497	1058	0	0
SRR22586062	PRJNA910346	SAMN32123559	3.67 G	SRP412145	3832	3239	593	0	0
SRR27211034	PRJNA1050165	SAMN38726487	436.05 M	SRP478043	147	0	147	0	0

## Data Availability

Nucleotide sequence data reported are available in the Third-Party Annotation Section of the DDBJ/ENA/GenBank databases under the accession numbers TPA: BK067724-BK067741. The virus sequences are also included as Appendix A for this submission.

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
