# Peer review of "RNA Virus Discovery Sheds Light on the Virome of a Major Vineyard Pest, the European Grapevine Moth (*Lobesia botrana*)"

_viruses, 2025, doi:10.3390/v17010095_

Round 1

Reviewer 1 Report

Comments and Suggestions for Authors

The grapevine moth, Lobesia bortrana is a dangerous pest of grapevine worldwide. It is extensively studied all over the world. Viruses are among important pathogens of insects, contributing to their population dynamics and biocontrol applications. However, a search in open bibliographic databases, such as Google Scholar, shows very limited number of papers devoted to viruses of L. botrana. The present paper provides a robust study of the grapevine moth virome utilizing bioinformatics to dig the available high-throughput sequencing metatranscriptomic data, an approach reaching out the actively expressed viruses (and leaving those just silently present in the host DNA). Meanwhile, the analyzed dataset includes 29 individual samples with varied history to empower the completeness of information. It perfectly fits the journal scope and will be interesting to the scientists who do similar research.

The paper is well organized and clearly written. I noticed only several minor flaws, listed below. The paper can be published after the respective corrections.

L21: suggesting … potential ecological roles – which roles?

L22: suggesting … host range – what do you mean?

L23: encoding proteins – what do you mean

L25-26: this family – which family?

L28: biological significance of this moth virus and grapevine – what do you mean?

L319-320: placed this virus within the Lepidoptera – sounds confusing

L397-404: This part better suits discussion

L515-516: first sobemo-like virus identified in moth - do you mean Lepidoptera?

Grammar issues:

Past and present tenses are sometimes mixed within a paragraph.

L21: a potential ecological roles – singular article misuse

L26-27: applications of these viruses on the L. botrana … – preposition misuse

L150: the only “segment” here not capitalized

L469: lepidopteran as its natural hosts – a mixture of singular and plural nouns

L496: it should be explored the possibility = the possibility should be explored

Author Response

The present paper provides a robust study of the grapevine moth virome utilizing bioinformatics to dig the available high-throughput sequencing metatranscriptomic data, an approach reaching out the actively expressed viruses (and leaving those just silently present in the host DNA). Meanwhile, the analyzed dataset includes 29 individual samples with varied history to empower the completeness of information. It perfectly fits the journal scope and will be interesting to the scientists who do similar research.

The paper is well organized and clearly written. I noticed only several minor flaws, listed below. The paper can be published after the respective corrections.

We thank reviewer #1 for taking the time to thoroughly assess our MS and provide suggestions which improved the MS.

L21: suggesting … potential ecological roles – which roles?

We deleted this sentence to avoid over assumptions

L22: suggesting … host range – what do you mean?

We deleted this sentence to avoid over assumptions

L23: encoding proteins – what do you mean

We refer to the proteins encoded by the Lobesia solemo-like virus

L25-26: this family – which family?

We refer to the Phasmaviridae family

L28: biological significance of this moth virus and grapevine – what do you mean?

we added “in the grapevine host” to clarify the sentence

L319-320: placed this virus within the Lepidoptera – sounds confusing

We refer that the Lobesia cypovirus was placed within the lepidoptera-associated genus, Cypovirus; we added a , to avoid any confusion

L397-404: This part better suits discussion

We believe that this paragraph is clearly written  

L515-516: first sobemo-like virus identified in moth - do you mean Lepidoptera?

We meand that is the first sobemo-like virus identified in moths

L21: a potential ecological roles – singular article misuse

This sentence was deleted (see comment above)

L26-27: applications of these viruses on the L. botrana … – preposition misuse

corrected (on was replaced with in)

L150: the only “segment” here not capitalized

corrected

L469: lepidopteran as its natural hosts – a mixture of singular and plural nouns

corrected

L496: it should be explored the possibility = the possibility should be explored

corrected

Reviewer 2 Report

Comments and Suggestions for Authors

Considered MS aim to analyze the presence of viral RNA in the big data results got using OMICS technique. My profile is not bioinformatics however I collaborate with bioinformatics tightly as well as I work with both RNA and DNA viruses so I can estimate the ideology of MS and basic results of the MS rather then technical details.  In general I think it is great idea to analyze viromes of host species, which provide scientific community by new information without significant expenses for working in wet lab. Moreover, microscopy, even the TEM, sometimes not informative enough to describe all viruses presented in insect host (i.e. Pavlushin et al., 2021, Virus research).  Of course approach decribed in submitted MS is not new but authors analyze new, economically important pest species that would be useful from applied point of view. In general the study looks solid and author used appropriate methods. To my opinion this MS will be interested the auditory of this journal as well as additional auditory of readers in pest management topic. I have some criticism which could improve the quality of this MS after that I think it could be suitable for the Viruses journal publication.

Major comments:

1) In method section authors provide careful factor description used for previous studies for getting raw data. I think this information confuse reader and stay behind the context of current study. If author would like to compare  different treatments and discus the results of viromes comparison (possibly diversity or something else) in the discussion, then no questions. But in current iteration I think it is redundant description confusing readers. 

2) I miss to find the quantitative comparison of viral RNA presence between different viruses. It is known that cypovirus able to replicate in huge amount even this virus infect only gut thus this could explain dominant amount of RNA copies in the samples. And from this description appear following question: were individuals alive or dead when they used for nuclear acid extraction? It is important to  mention and focus on this moment the readers because it will effect the following discussion: if insects were alive (i.e. all described infection were in chronic/latent/asymptomic  forms that is widespread for insect viruses, see review Williams et al., 2017 in Frontiers in microbiology) is it logical to speculate from pest control point of view? This is only first step to search highly pontent viral strain (for example cypovirus) which will regulate the density of L. botrana

3) L 561-564 I think virome instead holobiome is more strict term in the context of this sentence. You could see here (Rumiantseva et al., 2024, Journal of invertebrate pathlogy)  that interaction between RNA viruses and other components of microbiome within same host could be more complicated.

Minor comments

If you want to speculate about the potentioal of cypovirus in pest control even alive individuals were used for sample preparation (in the article – the source of raw data), you could discuss the potential to use adjuvant in the tank mixtures which could significantly increase the potency of RNA viruses (Shapiro, Dougherty, 1994, Journal of Economic Entomology; Takatsuka  2020, Journal of invertebrate pathlogy; Martemyanov et al., 2023, Microbiology spectrum)

Fig. 3 put the group numbers for all cypoviruses used in the picture () as it mentioned in the text, i.e. Daphnis nerii cypovirus 23.

Author Response

In general the study looks solid and author used appropriate methods. To my opinion this MS will be interested the auditory of this journal as well as additional auditory of readers in pest management topic. I have some criticism which could improve the quality of this MS after that I think it could be suitable for the Viruses journal publication.

We thank reviewer #1 for taking the time to thoroughly assess our MS and provide suggestions which improved the MS.

1) In method section authors provide careful factor description used for previous studies for getting raw data. I think this information confuse reader and stay behind the context of current study. If author would like to compare different treatments and discus the results of viromes comparison (possibly diversity or something else) in the discussion, then no questions. But in current iteration I think it is redundant description confusing readers. 

we deleted the paragraph “The Lobesia botrana individuals sampled and used for virus discovery correspond to both larvae and adults, with RNA extracted from whole individuals or from pheromone glands, including larvae, hosted on Vitis vinifera cv Cabernet Sauvignon or cv Riesling during flowering or veraison developmental stage exposed to elevated (ca. 480 ppm) or ambient (ca. 400 ppm) CO2 levels, or individuals susceptible or resistant to insecticides, and from Turkey or Germany” included in Lines 69-74

2) I miss to find the quantitative comparison of viral RNA presence between different viruses. It is known that cypovirus able to replicate in huge amount even this virus infect only gut thus this could explain dominant amount of RNA copies in the samples. And from this description appear following question: were individuals alive or dead when they used for nuclear acid extraction? It is important to mention and focus on this moment the readers because it will effect the following discussion: if insects were alive (i.e. all described infection were in chronic/latent/asymptomic  forms that is widespread for insect viruses, see review Williams et al., 2017 in Frontiers in microbiology) is it logical to speculate from pest control point of view? This is only first step to search highly pontent viral strain (for example cypovirus) which will regulate the density of L. botrana

Figure 5 shows the RNA levels of each virus, while Table 1 shows virus RPM in each run analyzed. We do not know if insects were alive or dead when they were processed for ARN extraction; thus, we cannot delve into this point with certainty.

3) L 561-564 I think virome instead holobiome is more strict term in the context of this sentence. You could see here (Rumiantseva et al., 2024, Journal of invertebrate pathlogy) that interaction between RNA viruses and other components of microbiome within same host could be more complicated.

we replaced “holobiome” with virome

4) If you want to speculate about the potentioal of cypovirus in pest control even alive individuals were used for sample preparation (in the article – the source of raw data), you could discuss the potential to use adjuvant in the tank mixtures which could significantly increase the potency of RNA viruses (Shapiro, Dougherty, 1994, Journal of Economic Entomology; Takatsuka  2020, Journal of invertebrate pathlogy; Martemyanov et al., 2023, Microbiology spectrum)

Since we do not know if insects were alive or dead when they were processed for ARN extraction we consider that it is better to not discuss this point; which also escapes to the aim of the manuscript

5) Fig. 3 put the group numbers for all cypoviruses used in the picture () as it mentioned in the text, i.e. Daphnis nerii cypovirus 23.

We modified the figure accordingly

Reviewer 3 Report

Comments and Suggestions for Authors

The manuscript by Debat et al. describes a metagenomics study of the virome of European grapevine moth, Lobesia botrana, conducted on in-silico data extracted from 29 publicly available libraries in the NCBI Sequence Read Archive (SRA). This is an interesting project that presents novel information on four new viruses discovered in these datasets that belong to four different families of viruses. Based on the information extracted from these 29 datasets, the authors even attempted to estimate the expression profiles and prevalence of these four viruses across different grapevine cultivars, growth stages, and environmental conditions. It would have been better if the authors could experimentally confirm the existence of these viruses in L. botrana samples, but as an in-silico project it can be accepted as is. While technologies and approaches used seem adequate to the proposed objectives, the writing of the manuscript and overall organization may be improved.

The manuscript is recommended for publication once the issues raised are properly addressed.   

Specific points to address:

ll. 37-47 – This introduction seems too brief and too generic, providing no clues on where (geographically) this insect is economically important, and giving no reasons why the authors selected this specific insect pest for their analysis. Ideally, an introduction needs to pose several questions that may be addressed in Results and then answered in Discussion: look at the 6-fold length disparity between the introduction (0.5-page) and discussion (3-pages).

ll. 67-74 – This section is redundant and confusing, suggesting that the authors did the extraction of RNA themselves; everything between the first and the last sentence of this paragraph should be deleted. All the information on host and tissue specificity is discussed later in the Results.

ll. 354-355 – This looks like a mistake, see a description of a positive-strand RNA genome at line 508.

Fig. 1 – resolution of the image in my pdf copy is not sufficient to see all the details, perhaps some of the less important images could be removed from this figure, or moved to supplementary figures?

Author Response

This is an interesting project that presents novel information on four new viruses discovered in these datasets that belong to four different families of viruses. Based on the information extracted from these 29 datasets, the authors even attempted to estimate the expression profiles and prevalence of these four viruses across different grapevine cultivars, growth stages, and environmental conditions. It would have been better if the authors could experimentally confirm the existence of these viruses in L. botrana samples, but as an in-silico project it can be accepted as is. While technologies and approaches used seem adequate to the proposed objectives, the writing of the manuscript and overall organization may be improved.

We thank reviewer #2 for taking the time to thoroughly assess our MS and provide suggestions which improved the MS.

  1. 37-47 – This introduction seems too brief and too generic, providing no clues on where (geographically) this insect is economically important, and giving no reasons why the authors selected this specific insect pest for their analysis. Ideally, an introduction needs to pose several questions that may be addressed in Results and then answered in Discussion: look at the 6-fold length disparity between the introduction (0.5-page) and discussion (3-pages).

We improved the introduction section as suggested, adding more specific information about L. botrana to add more context (see Lines 38-65).

  1. 67-74 – This section is redundant and confusing, suggesting that the authors did the extraction of RNA themselves; everything between the first and the last sentence of this paragraph should be deleted. All the information on host and tissue specificity is discussed later in the Results.

This paragraph was deleted according to the suggestion of both reviewers.

  1. 354-355 – This looks like a mistake, see a description of a positive-strand RNA genome at line 508.

corrected, thanks for pointing out this mistake

Fig. 1 – resolution of the image in my pdf copy is not sufficient to see all the details, perhaps some of the less important images could be removed from this figure, or moved to supplementary figures?

We uploaded in the figures and tables section  the original figure files, which has a high-quality resolution so the reviewer can see al details. Nevertheless, in the word file of the manuscript, the quality of Fig.1is good. 

Round 2

Reviewer 2 Report

Comments and Suggestions for Authors

Authors improve the previous version which looks almost suitable for the publication. I would attract the attention of authors to couple comments which will help reader (with different specialization) easy to consume the information and apply it for corresponding area. So after minor revision the MS could be published in the journal.

My comment#2

The first part of my comment was addressed to the absence of the DISCUSSION about presented results. Possibly the question was not clear formulated, sorry. I meant if there is the result about the quantity of viral RNAs it should be discussed in the  corresponded section of the MS. I also suggest the direction of the discussion in my previous comment. Otherwise why you put it in the main body of MS results instead supplementary.

Regarding the second part of the comment – I understand the reply and see my recommendation below.

My comment #4. I think there is the necessity to include your description in the text of discussion. i.e. “We used raw data obtained by another researchers and we do not know were larvae used for RNA extraction infected by acute viral infection (with morphology symptoms) or chronic, without symptoms. Since some of revealed viruses/strains may form chronic infection (for example CPVs) the result of our study is restricted only by the demonstration of the presence of potentially entomopathogenic viruses in Lobesia botrana. From more convincing recommendation to consider the CPV as potential insecticide additional studies (which is behind of the main idea of current study) is needed.” Possibly such type of speculation is redundant, but when I read the introduction, especially in the last iteration, this discussion will return the results of the study in the initial justification. Otherwise there is a gap between justification and the direction in results discussion. This is not obligate comment but I call authors to think abut it.

Other parts of MS are ok for me. No need to return to me improved version I think editor will able to make final decision without additional round of reviewing

Author Response

Authors improve the previous version which looks almost suitable for the publication. I would attract the attention of authors to couple comments which will help reader (with different specialization) easy to consume the information and apply it for corresponding area. So after minor revision the MS could be published in the journal.

We thank again the reviewer for taking the time to thoroughly assess our MS and provide suggestions which improved the MS.

The first part of my comment was addressed to the absence of the DISCUSSION about presented results. Possibly the question was not clear formulated, sorry. I meant if there is the result about the quantity of viral RNAs it should be discussed in the  corresponded section of the MS. I also suggest the direction of the discussion in my previous comment. Otherwise why you put it in the main body of MS results instead supplementary.

A sentence that reads as “As expected, LbCPV was the most consistently and abundantly expressed virus across all the libraries analyzed. It is well known that, in general, cypoviruses infect midgut epithelial cells of insect larvae and are able to replicate in huge amount causing chronic rather than lethal diseases in infected insect hosts [25]. Therefore, among the lobesia-associated viruses identified in our study, LbCPV could be a promising agent to manage L. botrana populations.” was added in Lines 463-468.

My comment #4. I think there is the necessity to include your description in the text of discussion. i.e. “We used raw data obtained by another researchers and we do not know were larvae used for RNA extraction infected by acute viral infection (with morphology symptoms) or chronic, without symptoms. Since some of revealed viruses/strains may form chronic infection (for example CPVs) the result of our study is restricted only by the demonstration of the presence of potentially entomopathogenic viruses in Lobesia botrana. From more convincing recommendation to consider the CPV as potential insecticide additional studies (which is behind of the main idea of current study) is needed.” Possibly such type of speculation is redundant, but when I read the introduction, especially in the last iteration, this discussion will return the results of the study in the initial justification. Otherwise there is a gap between justification and the direction in results discussion. This is not obligate comment but I call authors to think abut it.

A paragraph in the discussion (Lines 592-599) that reads as “Nevertheless, as we used raw data obtained by another researchers and we do not know if the larvae used for RNA extraction was infected by acute viral infection (with morphological symptoms) or chronic, without symptoms. Since some of revealed viruses/strains may form chronic infection (for example CPVs) the result of our study is restricted only by the demonstration of the presence of potentially entomopathogenic viruses in Lobesia botrana. From more convincing recommendation to consider the CPV as potential insecticide, additional studies (which is behind of the main idea of current study) are needed” was included in the manuscript.